# Chinese Version of the Vision-Related Quality of Life (NEI-VFQ-25) among Patients with Various Ocular Disorders: A Pilot Study

**DOI:** 10.3390/medicina58050602

**Published:** 2022-04-27

**Authors:** Jiahn-Shing Lee, Ken-Kuo Lin, Chiun-Ho Hou, Pei-Ru Li, Lai-Chu See

**Affiliations:** 1Department of Ophthalmology, Chang Gung Memorial Hospital at Linkou, Chang Gung University, Taoyuan City 333, Taiwan; leejsh@cgmh.org.tw (J.-S.L.); d12093@cgmh.org.tw (K.-K.L.); chiunho@cgmh.org.tw (C.-H.H.); 2Department of Public Health, College of Medicine, Chang Gung University, Taoyuan City 333, Taiwan; peili0506@cgmh.org.tw; 3Biostatistics Core Laboratory, Molecular Medicine Research Center, Chang Gung University, Taoyuan City 333, Taiwan; 4Division of Rheumatology, Allergy and Immunology, Chang Gung Memorial Hospital at Linkou, Taoyuan City 333, Taiwan

**Keywords:** vision-related quality of life, refractive error, keratoconus, senile cataract, age-related macular degeneration

## Abstract

*Background and Objectives*: Subjective visual function is currently becoming an increasing appreciation in assessing the health-related quality of life. This study aimed to assess the vision-related quality of life (VRQOL) among patients with refractive errors, keratoconus, senile cataract, and age-related macular degeneration (AMD) using the Chinese version of the National Eye Institute Visual Function Questionnaire 25 (NEI-VFQ-25). *Materials and Methods*: The questionnaire of NEI-VFQ-25 was filled out in a clinical setting or by telephone/mail. Univariate and multivariate analyses were used to determine which factors are associated with the NEI-VFQ-25. *Results*: From June 2018 to January 2019, 28 patients with refractive error, 20 patients with keratoconus, 61 with senile cataracts, and 17 with AMD completed the questionnaire NEI-VFQ-25. There were significant differences in the NEI-VFQ-25 subscale of general vision (*p* = 0.0017), ocular pain (*p* = 0.0156), near activities (*p* = 0.0002), vision-specific social functioning (*p* = 0.007), vision-specific mental health (*p* = 0.0083), vision-specific dependency (*p* = 0.0049), color vision (*p* < 0.0001), peripheral vision (*p* = 0.0065), and total score (*p* < 0.0001) among four disease groups, respectively. The multiple linear regression revealed that the best-corrected visual acuity (BCVA) and disease group were important factors of the total NEI-VFQ-25. After adjusting for BCVA, patients with AMD had a worse total NEI-VFQ-25 score than patients with refractive error, keratoconus, or senile cataracts. *Conclusions*: Among the patients with four ocular disorders and a broad vision spectrum from normal, partial sight, low vision to legal blindness, the BCVA of their better eye was the most important factor in the VRQOL.

## 1. Introduction

Vision is the most widely recognized perceptual modality. Our impressions about the world and our memories are predominantly based on sight [1]. Our eyeball consists of two refractive tissues (cornea and lens) and one photoreceptor tissue (the macula in its center). Refractive errors due to abnormal axial length or sphero-cylindrical cornea, senile cataract due to lens opacity, and age-related macular degeneration (AMD) are the three leading causes of visual impairment in the world; and keratoconus due to irregular shape of the cornea is another representative ocular disorder.

In the past few decades, subjective visual function has seen an increasing appreciation of assessing the health-related quality of life (HRQOL), which evaluates the effect of a disease on both the patients’ biological status and the intricate sequelae of their life. In ophthalmology, tools to measure visual quality and vision-related quality of life (VRQOL) have been continuously developed and are increasingly adopted. Numerous questionnaires have been made to assess VRQOL [2,3,4,5]. There are two broad categories of VRQOL instruments: generic instruments for different ocular disorders and disease-specific instruments. The National Eye Institute Visual Function Questionnaire-25 (NEI-VFQ-25), a generic instrument, has been widely studied. It evaluates the quality of vision and the VRQOL and is suitable for analyzing many eye diseases, such as glaucoma, AMD, diabetic retinopathy, cytomegalovirus retinitis, and senile cataracts. There are many translated versions of the NEI-VFQ-25, including a Taiwan Chinese version whose validity and reliability are equivalent to the original version [6,7].

Refractive errors without correction are the principal cause of poor vision worldwide and can severely affect VRQOL [8,9]. The correction of refractive errors using spectacles or contact lenses is the most cost-effective intervention [8]. To avoid the hindrance of spectacles or contact lenses, many people may seek surgical correction of their refractive errors. Correcting myopia using SMILE (small incision lenticule extraction), LASIK (laser in situ keratomileusis), or PRK (photorefractive keratectomy) has been found to produce higher VRQOL scores as compared with wearing spectacles or contact lenses [10,11].

Keratoconus is a gradually and unevenly developed noninflammatory disorder with steeping, thinning, and cornea scarring. These alterations in the cornea result in irregular astigmatism and subsequent vision impairment [12]. Mild keratoconus can be treated by wearing spectacles or soft contact lenses. Rigid or special contact lenses are considered when vision is not correctable by spectacles and patients become symptomatic. Approximately 11% to 27% of keratoconus patients will deteriorate to a visual state that the contact lenses cannot correct, and a corneal surgery or transplantation is needed [13].

Senile cataracts are a leading cause of visual impairment [14]. Cataract surgery is the most commonly performed surgery in the eye clinic with high satisfaction [15]. Traditional eye exams, such as visual acuity and contrast sensitivity, are the most-accepted assessments in practice for cataracts and their management. The NEI-VFQ-25 is another effective measure of patients’ visual capability and VRQOL pre-and post-operatively. Both types of evaluation should be considered simultaneously to measure visual function [16].

Age-related macular degeneration (AMD) is another leading cause of visual impairment in people aged 65 or more [17]. The vision loss in patients with AMD mainly affects their ability to perform daily activities, causing distress and social isolation [18,19]. The NEI-VFQ-25 had been applied among AMD patients and has demonstrated correlations with a spectrum of vision measurements and a daily function measure [20]. Interestingly, there were substantial differences in the NEI-VFQ-25 scores among AMD patients from different countries [21].

This study aimed to assess the VRQOL using the Chinese version of the National Eye Institute Visual Function Questionnaire 25 (NEI VFQ-25) among patients with refractive errors, keratoconus, senile cataract, and AMD in Taiwan. The inclusion of these four different eye diseases, which have different effects on sight -from minor to low vision, provides a wide range of NEI-VFQ-25 scores.

## 2. Materials and Methods

### 2.1. Study Design

A questionnaire survey was performed in a supervised clinical setting and via mail. A voucher of TWD 50 was given to those who completed the questionnaire. The study was conducted in accordance with the Declaration of Helsinki. We got approval from the Institutional Review Board (IRB) of Chang Gung Medical Foundation (201601721B0: approved on 19 December 2016, 201601721B0C601: approved on 31 May 2018, 201601721B0C602: approved on 8 October 2018). All subjects gave their written informed consent for inclusion before they participated in the study.

From May 2018 to January 2019, 147 participants filled out the questionnaire NEI-VFQ-25. After excluding 21 (missing BCVA), 126 patients (28 with the refractive error problem, 20 patients with keratoconus, 61 with senile cataract, and 17 with AMD) were eligible for this study. Most cases were filled during their clinical visits, and 6 were returned by mail.

### 2.2. Variables of Interest

The NEI-VFQ-25 assesses the self-reported, vision-targeted health status of people with chronic eye diseases. The NEI-VFQ-25 has been used to evaluate the subjective visual function and health-related quality of life between people free of eye disease and those with eye disease and detect changes after various intraocular procedures such as contact lens wearing or surgery [7,22,23,24,25,26,27,28]. The NEI-VFQ-25 contains 25 questions: general health, general vision, ocular pain, distance, near tasks, dependency on others, role limitation, mental health, social function, driving, peripheral vision, and color vision difficulty. The answer is converted into a 100-point scale for each question, with 100 being the best and 0 the worst. One or more questions are specific to each subscale; therefore, the subscale score is the average of one or more specific questions. The Chinese version of the NEI-VFQ-25 questionnaire was used [6].

The best-corrected visual acuity (BCVA) was obtained by reviewing their most recent medical records. Because of its geometric nature, the BCVA was converted to the logarithm of the minimum angle of resolution (logMAR) chart before averaging and converted back to decimal acuity [29] as below:logMAR = −log (decimal acuity)(1)
Decimal acuity = antilog (−LogMAR) = 10^−LogMAR^(2)

### 2.3. Statistical Analysis

Descriptive statistics such as the mean, standard deviation and frequency were used. The frequency of each item in the NEI-VFQ-25 was listed. Because the rating of some items in the questionnaire was not consistent, we reversed the rating for those items and then computed a sum of NEI-VFQ-25. The Pearson correlation coefficient (r) and scatter plot were used to explore the relationship among continuous variables. Chi-square tests, ANOVA, or Kruskal–Wallis tests were conducted to examine which factors (demographic variables including age, sex, and BCVA, or various ocular disorder) were associated with the NEI-VFQ-25, univariately. Multiple comparisons by either Scheffe’s method or Bonferroni correction were conducted to determine which group was different when ANOVA or Kruskal–Wallis tests reached significance. Multiple linear regression with forward selection was used to determine which factors were multivariately associated with the NEI-VFQ-25. The significance level of this study was 0.05.

## 3. Results

Among 126 patients, patients with refractive error and keratoconus were younger (mean age of 38 years old) than patients with senile cataracts and AMD (mean age of 68 and 70 years old, respectively) (*p* < 0.0001). There were more female patients with AMD (64.7%) and fewer female patients with keratoconus (40%), but the statistical difference in sex between the four disease groups did not reach significance (*p* = 0.4936). The BCVA was worse in patients with AMD for both the better eye and the worse eye than in the other three groups (*p* < 0.0001) (Table 1).

There were significant differences in the NEI-VFQ-25 subscale scores of general vision (*p* = 0.0017), ocular pain (*p* = 0.0156), near activities (*p* = 0.0002), vision-specific social functioning (*p* = 0.007), vision-specific mental health (*p* = 0.0083), vision-specific dependency (*p* = 0.0049), color vision (*p* < 0.0001), peripheral vision (*p* = 0.0065), and total score (*p* < 0.0001) among the four disease groups, respectively. There was no difference in general health (*p* = 0.3555), distance activities (*p* = 0.1304), vision-specific role difficulties (*p* = 0.1734), driving (*p* = 0.7426) among the four disease groups, respectively (Table 2). It should be noted that only 58 subjects [refractive error (*n* = 19), keratoconus (*n* = 12), senile cataract (*n* = 25), AMD (*n* = 2)] filled out the question about driving. Despite the low score of the driving question in those with AMD, the insignificance may be due to a small number of these patients (*n* = 2).

Table 3 summarizes the univariate analysis of the total score of the NEI-VFQ-25. The total score of the NEI-VFQ-25 significantly differed in the four disease groups (*p* = 0.0002), three age groups (*p* = 0.0466), and with BCVA (*p* < 0.0001 for both the better eye and the worse eye). The AMD patients had the worst total NEI-VFQ-25 score, significantly different from the other three eye disease groups (refractive error, keratoconus, and senile cataracts). There was no significance in the total NEI-VFQ-25 score among these three eye disease groups. Patients of 60 years or older had a significantly worse total NEI-VFQ-25 score. The total NEI VFQ-25 score was negatively correlated with age (r = −0.22) (Figure 1). The total NEI VFQ-25 score decreased with the BCVA of logMAR for both the better eyes and the worse eyes increased (r = −0.46 and −0.38, respectively) (Figure 2 and Figure 3).

The multiple linear regression reveals that BCVA and disease group were important to the total NEI-VFQ-25 score. The total R^2^ was 24.8%. BCVA was first chosen in the model with an R^2^ of 20.8%, and ocular disease grouping further explained an R^2^ of 4.0%. When BCVA increases one unit of logMAR for the better eye, the total NEI-VFQ-25 score decreases by 18.3 units after adjusting for the disease group. Regarding the disease group, patients with AMD scored 9.8 units of the total NEI-VFQ-25 lower than patients with cataracts after adjusting for the BCVA of logMAR for better eyes (Table 4).

## 4. Discussion

This study analyzed patients with four common eye diseases and a broad vision range from normal sight, partial sight, and low vision to legal blindness. There were significant differences in the NEI-VFQ-25 subscale scores of general vision, ocular pain, near activities, vision-specific social functioning, vision-specific mental health, vision-specific dependency, color vision, peripheral vision, and total score among the four disease groups, respectively. The multiple linear regression reveals that the BCVA of the better eye and the disease group were important in the total NEI-VFQ-25 score. After adjusting for BCVA, patients with AMD had a worse total score of NEI-VFQ-25 than patients with refractive error, keratoconus, or senile cataracts.

Similar to our finding, Lin et al. reported that visual impairment was associated with a lower VRQOL, as assessed by NEI-VFQ-25 in Taiwanese patients [30]. Patients in Lin’s study were aged 40 years or more (with a median age of 74.0 years) and a BCVA of 20/40 or worse in the better eye. They concluded that a history of heart disease, arthritis, and eye diseases, such as AMD or diabetic retinopathy, had significant adverse effects on the patient’s HRQOL. Every unit increase in logMAR decreased the total NEI-VFQ-25 globally. According to this analysis, patients’ HRQOL was improved by each unit increase in BCVA.

Regarding visual ability, only the BCVA of the better eye had a significant effect on VRQOL. While, in our study, the BCVA increased one unit of logMAR for the better eye, the total score of the NEI-VFQ-25 decreased by 18.3 units after being adjusted for the disease group. Regarding the disease group, patients with AMD had a total score of NEI-VFQ-25 9.8 units lower than patients with cataracts after adjusting for the BCVA of logMAR for better eyes. The BCVA is determined by objective refraction, such as via retinoscope or auto-refractometer, and subjective refraction. The auto-refractometric exam is rapid and provides fewer deviations than the subjective method with normal BCVA [31,32]. However, in diseased eyes, such as those with keratoconus or cataracts, there are high errors of auto-refractometric readings or even no readings. The determination of BCVA in such eyes is laborious and needs skill. Therefore, a flexible methodology to determine visual function may be helpful.

Visual impairment is vision reduction not corrected by glasses or contact lenses. The World Health Organization uses the vision in the better eye to classify visual impairments with the best possible glasses correction. For example, ‘blindness’ is defined as a visual acuity of less than 3/60, or a corresponding visual field loss to less than 10°, in the better eye with the best possible correction [33]. This definition was set in 1972, lasting for nearly a half-century [34]. Since our results, consistent with many previous reports, revealed that the ‘BCVA of the better eye’ was the most crucial factor in correlating the visual life quality, it has still been a good, though old, parameter for the definition classification of visual impairments [34]. It was also reported that having one eye with good vision in unilateral moderate to severe visual impairment is enough to have a high HRQOL [35]. Besides, after adjusting for vision and other comorbidities, many measures of HRQOL were found to be less or not related to many senile ocular diseases [36].

Even adjusted for the BCVA, the ocular disease grouping possessed an extra 4% of the variability of the NEI-VFQ-25. Patients with AMD had a significantly higher NEI-VFQ-25 score than patients with senile cataracts, although these two patient groups had similar mean ages. The worse NEI-VFQ-25 score in AMD compared with other ocular diseases is well recognized [37]. AMD confers significant functional difficulty among adults with sociodemographic characters influencing dysfunction, stressing the worth of other measures to supplement Snellen’s vision test in assessing visual characteristics. In an aging society and with the increasing prevalence of AMD, health care providers should be aware of the functional burden of AMD and recognize those at a higher risk of disability [38]. We suggest that eye-care providers be aware of this research evidence and refer patients for low vision rehabilitation or social counseling services. There are two types of AMD, dry (or atrophic) and wet (or neovascular). Dry AMD usually progresses slowly over many years, and wet AMD, a less common type of late AMD, frequently causes faster visual loss. Many studies of VRQOL have focused on wet AMD, and one investigated the natural progression of AMD and found different scores between dry and wet AMDs, suggesting that future studies should consider separating these phenotypes [39]. Moreover, females and those with a higher baseline VRQOL were more likely to have a steep decline in VRQOL following AMD progression. Therefore, it is noteworthy that the substantial differences in NEI-VFQ-25 scores among AMD patients of different countries may be related to the different proportions of AMD types, stages, patient age, and gender [21].

The prevalence of cataracts increases with aging, and half of the population have cataracts by age 75. They may cause gradual vision loss and are an essential issue in public health. Surgery is the only means to restore vision in the vision-threatening cataract. Cataract surgery aims to improve the patient’s visual function and, eventually, quality of life. However, two commonly used VRQOL questionnaires, VF-14 (visual function index-14) and NEI-VFQ-25, were found to have deficiencies with regard to several critical psychometric properties, including unidimensionality, targeting, and differential item functioning [40]. Ceiling defects were also observed on nine of the twelve subscales in the NEI-VFQ-25.

Further studies are required to confirm such deficiencies and determine whether they occurred only in cataract patients or in other cases of different visual problems. There is another trend in developing a different method for the evaluation of visual function, such as the real-life vision test (RLVT) [16]. The use of the RLVT combined with clinical and self-survey methods provides a new approach to ascertaining the impact of cataract surgery on patients’ overall VRQOL.

Patients with refractive error and keratoconus were two young groups studied here. The NEI-VFQ-25 score was lower for patients with keratoconus than those with refractive error. Keratoconus is usually detected in the teenage years or the twenties. The changes in the cornea shape may occur over several years but at a more rapid rate in some younger patients. As with the stress of facing degenerative and disabling diseases, young adults with keratoconus might experience anxiety about the possibility of their disease progression, future surgery, and vision loss. They also suffer the burden of frequent hospital visits.

Because patients with moderate keratoconus are uncorrectable by spectacles owing to their distorted cornea, they may suffer additional anxiety in wearing contact lenses which is the only corrective method. However, such contact lenses may not be tolerated for a whole-day-long wearing due to several reasons such as irritation, and then patients have to live with blurring vision occasionally. Thus, it may be appropriate to use a comprehensive tool to examine these patients [41,42]. The VRQOL in patients with keratoconus has a lifetime impact, affecting vocation, livelihood, and social integration. The vision impairment and its related symptoms in keratoconus patients ranging from mild to severe impairment are related to the disease’s stage and have different impacts on HRQOL. Daily activities and emotional welfare have also been shown to reduce as the severity of KC increases.

Refractive error without correction is associated with a decreased VRQOL and visual dysfunction [8,9]. Such association has been found in various populations using different functional assessment tools, such as the NEI-VFQ-25. The spectacle correction of refractive error improving quality-of-life has been reported. However, in a Latino population study, the correction of any refractive error, including myopia, hyperopia, or astigmatism, is not sufficient to restore the function of far vision to a level of those without refractive error [43]. It has been shown that NEI-VFQ-25 scores are affected by demographic factors, including age, gender, level of socialization, income, education level, and comorbidities such as diabetes and hypertension [44]. This Latino study also found that these factors are associated independently with visual function at both far and near distances regardless of the types of refractive error. Although the NEI-VFQ-25 is vision-specific, it is related to other parameters such as cultural factors. Therefore, it is crucial to adjust for these and other possible factors when measuring vision dysfunction.

Our study’s key strength was using a standardized instrument among patients with four common eye diseases and a broad vision range from normal sight, partial sight, and low vision to legal blindness to assess their VRQOL. Our study emphasized the importance of both VRQOL and BCVA. The BCVA is a routine exam in an eye clinic, and VRQOL is an alternate for other medical personnel such as social workers. The BCVA is recommended to assess patients’ visual function, and VRQOL for health-related life quality.

On the other hand, our study has limitations. First, it has been reported that depression is associated with self-reported functional status, but we did not collect depression status in this study [45]. Similarly, about 40% of patients reported feeling moderately or extremely anxious or depressed among Taiwanese patients with visual impairment [30]. Second, the lack of a definition for the disease severity of these ocular disorders presents a challenge for inter- or intra-disease comparison in the future. The type and stage of different ocular diseases were not recorded either. Hence, we cannot examine their effect on VRQOL. Third, this study did not include another visual impairment category, visual field defects such as glaucoma cases. Fourth, the sample sizes for the four ocular diseases were small, especially for patients with refractive error, KC, and AMD. For KC, the male: female ratio (60:40) in this study was similar to that of the KC population in Taiwan (59.7:40.8), but the mean age in this study (37.9 years) was older than that of the KC population in Taiwan (29.7 years) [46]. For senior cataracts, the mean age (68.3 years) and the male: female ratio (45.9:54.1) in this study were similar to those of the cataract population in Taiwan (69.6 years, 46.5:53.5) [47]. There is a high prevalence of myopia in Taiwan for refractive errors, the female slightly higher than or similar to the male as in this study [48]. For AMD, the mean age in this study (70.0 years) was similar to that of the AMD population in Taiwan, but the male: female ratio (35.3:65.7) in this study was different from that of the AMD population (equal gender distribution) in Taiwan [49]. We recognize that this study’s patients with the four ocular diseases do not represent the population. Hence, we suggest that more patients with the four ocular diseases are needed. Furthermore, patients with low vision (decimal acuity 0.05 or less) can help to improve the estimate of regression coefficients in Table 4.

## 5. Conclusions

In conclusion, among the patients with four ocular disorders and a broad vision spectrum from normal, partial sight, low vision to legal blindness, the BCVA of their better eye was the most crucial factor in correlating the VRQOL. However, even adjusted for BCVA, patients with AMD had a significantly worst VRQOL, and low vision rehabilitation or social counseling services are suggested.

## Figures and Tables

**Figure 1 medicina-58-00602-f001:**
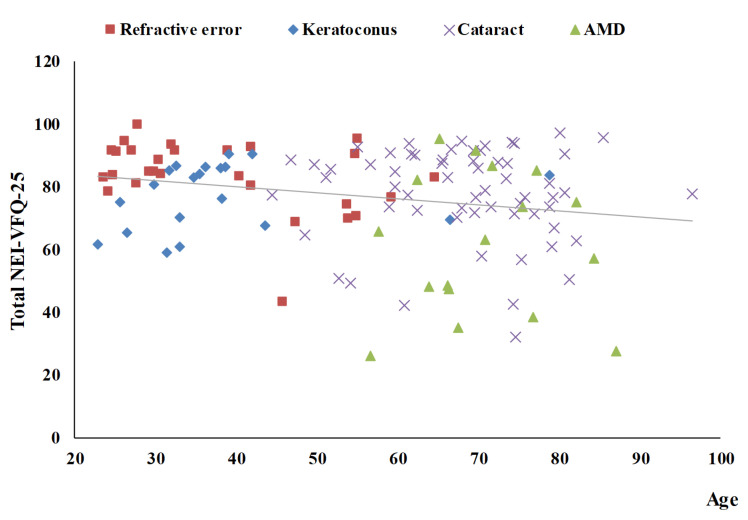
The scatter plot between total NEI-VFQ-25 score and age (r = −0.22, *p* = 0.0118).

**Figure 2 medicina-58-00602-f002:**
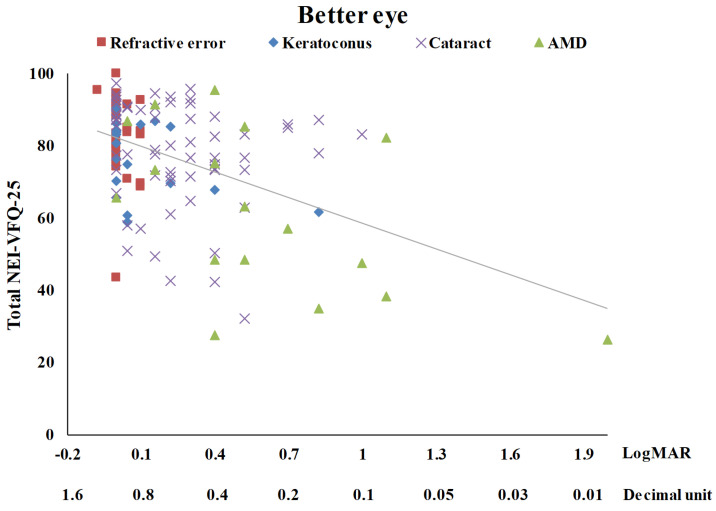
The scatter plot between total NEI-VFQ-25 score and the best-corrected visual acuity for the better eye (r = −0.46, *p* < 0.0001).

**Figure 3 medicina-58-00602-f003:**
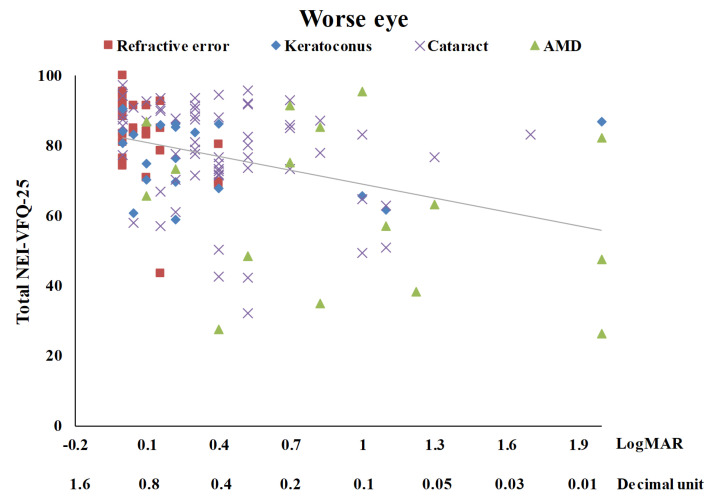
The scatter plot between total NEI-VFQ-25 score and the best-corrected visual acuity for the worse eye (r = −0.38, *p* < 0.0001).

**Table 1 medicina-58-00602-t001:** Demographic characteristics among four study groups with different ocular disorders (*n* = 126).

	Refractive Error (*n* = 28)	Keratoconus (*n* = 20)	Cataract (*n* = 61)	AMD (*n* = 17)	*p*-Value
Age (years)					<0.0001 ^1^
20–39	16 (57.1%)	16 (80.0%)	0	0	
40–59	11 (39.3%)	2 (10.0%)	14 (23.0%)	2 (11.8%)	
≥60	1 (3.6%)	2 (10.0%)	47 (77.0%)	15 (88.2%)	
Mean ± SD	38.2 ± 12.8 ^A^	37.9 ± 13.1 ^A^	68.3 ± 10.6 ^B^	70.0 ± 8.8 ^B^	<0.0001 ^2^
Sex					0.4936 ^3^
Male	14 (50.0%)	12 (60.0%)	28 (45.9%)	6 (35.3%)	
Female	14 (50.0%)	8 (40.0%)	33 (54.1%)	11 (64.7%)	
Visual acuity (corrected) (logMAR/decimal unit)					
Better eye	0.02 ± 0.04/0.95 ± 0.4 ^A^	0.10 ± 0.20/0.79 ± 2.0 ^AC^	0.25 ± 0.24/0.56 ± 2.4 ^C^	0.60 ± 0.49/0.25 ± 4.9 ^B^	
Worse eye	0.09 ± 0.12/0.82 ± 1.2 ^A^	0.34 ± 0.49/0.46 ± 4.9 ^AC^	0.43 ± 0.35/0.37 ± 3.5 ^C^	0.91 ± 0.63/0.12 ± 6.3 ^B^	

^1^ Chi-square test for trend; ^2^ ANOVA; ^3^ Chi-square test; ^A, B, C^ Multiple comparisons: Different letters represent a significant difference between groups, and the same letters represent no difference between groups. AMD: age-related macular degeneration.

**Table 2 medicina-58-00602-t002:** NEI-VFQ-25 subscale scores and composite score (*n* = 126).

	Refractive Error (*n* = 28)	Keratoconus (*n* = 20)	Cataract (*n* = 61)	AMD (*n* = 17)	*p*-Value
General health	48.2 ± 26.3	55.0 ± 25.1	43.0 ± 26.3	42.6 ± 26.2	0.3555 ^1^
General vision	67.1 ± 17.4 ^AB^	69.0 ± 17.7 ^A^	54.7 ± 18.7 ^B^	50.6 ± 21.4 ^B^	0.0017 ^1^
Ocular pain	76.8 ± 17.6 ^AB^	65.6 ± 18.1 ^A^	80.5 ± 22.0 ^B^	74.3 ± 22.7 ^AB^	0.0156 ^1^
Near activities	91.4 ± 14.3 ^A^	91.2 ± 12.2 ^AC^	80.8 ± 20.0 ^BC^	64.1 ± 25.4 ^B^	0.0002 ^1^
Distance activities	86.2 ± 14.0	83.3 ± 14.6	81.8 ± 21.3	69.8 ± 23.1	0.1304 ^1^
Vision specific:					
Social functioning	95.5 ± 11.4 ^A^	93.8 ± 13.1 ^A^	90.6 ± 16.2 ^A^	72.8 ± 25.5 ^B^	0.0007 ^1^
Mental health	78.6 ± 18.0 ^A^	66.9 ± 23.2 ^AB^	74.1 ± 18.5 ^A^	54.8 ± 26.7 ^B^	0.0083 ^1^
Role difficulties	74.6 ± 22.2	63.1 ± 20.1	66.6 ± 28.8	56.6 ± 31.9	0.1734 ^1^
Dependency	95.7 ± 8.0 ^A^	83.3 ± 19.3 ^AB^	84.6 ± 23.0 ^AB^	56.9 ± 40.0 ^B^	0.0049 ^1^
Driving	80.3 ± 15.5	67.4 ± 33.8	73.0 ± 29.1	43.8 ± 61.9	0.7426 ^1^
Color vision	97.3 ± 10.4 ^AB^	98.8 ± 5.6 ^A^	95.9 ± 12.2 ^AB^	78.3 ± 20.8 ^B^	<0.0001 ^1^
Peripheral vision	90.2 ± 18.4 ^A^	80.0 ± 25.1 ^AB^	85.7 ± 19.6 ^AB^	65.0 ± 28.0 ^B^	0.0065 ^1^
Total (25-item composite)	83.7 ± 11.3 ^A^	77.5 ± 10.4 ^A^	77.8 ± 14.9 ^A^	61.6 ± 22.7 ^B^	<0.0001 ^2^

^1^ Kruskal–Wallis test; ^2^ ANOVA; ^A, B, C^ Multiple comparisons: Different letters represent a significant difference between groups, and the same letters represent no difference between groups. AMD: age-related macular degeneration. NEI-VFQ-25: National Eye Institute Visual Function Questionnaire 25.

**Table 3 medicina-58-00602-t003:** Univariate analysis of the total score of NEI-VFQ-25 (*n* = 126).

		Total (25-Item Composite)	
	*n* (%)	Mean ± SD	*p*-Value
Group			0.0002 ^1^
Refractive error	28 (22.2%)	83.7 ± 11.3 ^A^	
Keratoconus	20 (15.9%)	77.5 ± 10.4 ^A^	
Cataract	61 (48.4%)	77.8 ± 14.9 ^A^	
AMD	17 (13.5%)	61.6 ± 22.7 ^B^	
Age (years)		−0.2236^2^	0.0118 ^2^
20–39	32 (25.4%)	82.9 ± 10.1 ^A^	0.0466 ^1^
40–59	29 (23.0%)	75.6 ± 16.5 ^AB^	
≥60	65 (51.6%)	74.5 ± 17.7 ^B^	
Sex			0.9025 ^3^
Male	60 (47.6%)	77.1 ± 14.9	
Female	66 (52.4%)	76.7 ± 17.2	
Corrected visual acuity (logMAR)			
Better eye		−0.4556 ^2^	<0.0001 ^2^
Worse eye		−0.3757 ^2^	<0.0001 ^2^

^1^ ANOVA; ^2^ Pearson correlation; ^3^ Independent *t*-test; ^A, B^ Multiple comparisons: Different letters represent a significant difference between groups, and the same letters represent no difference between groups. AMD: age-related macular degeneration. NEI-VFQ-25: National Eye Institute Visual Function Questionnaire 25.

**Table 4 medicina-58-00602-t004:** Multivariate analysis of the total score of NEI VFQ-25 (*n* = 126).

	Regression Coefficient ± Standard Error	*p*-Value	Cumulative R^2^
Intercept	82.4 ± 2.2	-	-
Visual Acuity: better eye (logMAR)	−18.3 ± 5.0	0.0004	20.8%
Disease Group			24.8%
Refractive error	1.7 ± 3.4	0.6250	
Keratoconus	−3.1 ± 3.7	0.4054	
AMD	−9.8 ± 4.3	0.0226	
Cataract	reference	-	

AMD: age-related macular degeneration. NEI-VFQ-25: National Eye Institute Visual Function Questionnaire 25.

## Data Availability

The data that support the findings of this study are available on request from the corresponding author.

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
