# Peer review of "Chinese Version of the Vision-Related Quality of Life (NEI-VFQ-25) among Patients with Various Ocular Disorders: A Pilot Study"

_medicina, 2022, doi:10.3390/medicina58050602_

Round 1

Reviewer 1 Report

None.

Reviewer 2 Report

All comments had been answered

thank you

This manuscript is a resubmission of an earlier submission. The following is a list of the peer review reports and author responses from that submission.

Round 1

Reviewer 1 Report

Lee and colleagues performed a cross-sectional survey study among patients with refractive errors, keratoconus, senile 20 cataract, and age-related macular degeneration (AMD) using the Chinese version of the National Eye Institute Visual Function Questionnaire 25 (NEI-VFQ-25) to examine the vision-related quality of life (VRQoL). The study reports a significant difference among a number of parameters assessed using the NEI-VFQ-25 questionnaire and the different disease groups and concludes that best-corrected visual acuity (BCVA) of the better eye was the most important determinant of VRQoL score.

The manuscript is well written and clearly describes the instruments used in the methods. The study is performed in an ethical manner. The statistical methods utilized are appropriate.

Minor concerns and corrections -

Delete HRQOL in abstract – Line 19.

References line 222. “Many studies….”(no references given which studies)

As the authors have pointed out that age was highly variable in the study groups ranging from 20 to greater than 60 (p=0.000) in table 1 because of investigating diseases of the young (refractive error and keratoconus) and the aged (cataract and AMD). What happens when the investigators would include age in the multivariate analysis model (Table 4)?

The authors need to address the limitation of sample size in each disease group. Will increase in sample affect the results?

Reviewer 2 Report

The manuscript aims to assess the vision-related quality of life in patients with different ocular pathologies - refractive errors, keratoconus, cataract and AMD. The inclusions of different eye diseases that has different effect on sight - from minor to low vision,- provides wide range of NEI-VFQ-25 usage.

The manuscripts is well structured, has all the appropriate parts. Though most of the references are older that 5 year (5 out of  42 are within the last five years).

The Material and methods section lacks precise description of the participants included (what type of refractive errors, stage of keratoconus, AMD, cataract). The number of the participants is low considering the extent of the diseases and lack of information regarding the stage of the disease.

Results are presented clearly and the conclusions are supported by the results but it does not provide novel information to the field. The figures 1-3 lack some essential information - what type of correlation coefficient was used, does the correlation reach level of significance.

The editing of English language is required. There are some missing words in the text. 

Reviewer 3 Report

Thank you for allowing me to review this paper

  1. Indeed, we need to evaluate more than visual acuity. Having subjective visual function outcome is indeed of utmost importance to compare methods.
  2. This is a Chinese version of the NEI-VFQ-2, please add this to the title similar to https://pubmed.ncbi.nlm.nih.gov/28430336/
  3. The intro could be modified- from Human have ….to the part of the eyes anatomy.- keep it more scientific
  4. a Chinese version of the NEI-VFQ-2 had been published -ref 6
  5. Why did you stop in 2019, not till 2021?
  6. What is the aim of this study to evaluate which group had worse VRQOL? How does this help the patients?

You wrote:" This study aimed to assess the VRQOL, using the Chinese version of the National Eye Institute Visual Function Questionnaire 25 (NEI VFQ-25), among patients with refractive errors, keratoconus, senile cataract, and AMD in Taiwan."- please explain why to do this work? What is the motive?

  1. So you found the corrected vision is related to VRQOL- this is logical, no? you need good vision to have a good quality of vision, no?

Overall, significant work has been done here. However, please try to rephrase it more clinical or make it beneficial to the reader- what is the benefit of reading this paper? How will it change sometimes? What will be changed? How will it improve the field? How will it improve patients' life?

Round 2

Reviewer 3 Report

My comment (only one)   If BCVA is quick and routine and it predict the VRQOL   Why do we need to do a VRQOL? It’s Time consuming
